# Picture perfect science communication: How public audiences respond to informational labels in cinematic-style 3D data visualization

Eric Allen Jensen[1,2]*, Kalina Borkiewicz[2,3], Jill P. Naiman[4], Stuart Levy[2], Jeff Carpenter[2]

**1** Institute for Methods Innovation, Casper, Wyoming, United States of America, **2** National Center for Supercomputing Applications, University of Illinois Urbana-Champaign, Urbana, Illinois, United States of America, **3** University of Utah, Scientific Computing and Imaging Institute, Salt Lake City, Utah, United States of America, **4** Information School, University of Illinois Urbana-Champaign, Champaign, Illinois, United States of America

\* eric@methodsinnovation.org

**Data Availability Statement:** The dataset underpinning this paper have been published here: https://zenodo.org/records/13284016.

## Abstract

Visualizing research data can be an important science communication tool. In recent decades, 3D data visualization has emerged as a key tool for engaging public audiences. Such visualizations are often embedded in scientific documentaries screened on giant domes in planetariums or delivered through video streaming services such as Amazon Prime. 3D data visualization has been shown to be an effective way to communicate complex scientific concepts to the public. With its ability to convey information in a scientifically accurate and visually engaging way, cinematic-style 3D data visualization has the potential to benefit millions of viewers by making scientific information more understandable and interesting. Maximizing the effectiveness of 3D data visualization can benefit millions of viewers. To support a wider shift in this professional field towards more evidence-based practice in 3D data visualization to enhance science communication impact, we have conducted a survey experiment comparing audience responses to two versions of 3D data visualizations from a scientific documentary film on the theme of 'solar superstorms' (n = 577). This study was conducted using a single (with two levels: labeled and unlabeled), between-subjects, factorial design. It reveals key strengths and weaknesses of communicating science using 3D data visualization. It also shows the limited power of strategically deployed informational labels to affect audience perceptions of the documentary film and its content. The major difference identified between experimental and control groups was that the quality ratings of the documentary film clip were significantly higher for the 'labeled' version. Other outcomes showed no statistically significant differences. The limited effects of informational labels point to the idea that other aspects, such as the story structure, voiceover narration and audio-visual content, are more important determinants of outcomes. This study concludes with a discussion of how this new research evidence informs our understanding of 'what works and why' with cinematic-style 3D data visualizations for the public.

**Funding:** All authors worked on The Brinson Foundation grant, but the PI was KB. There was no grant number. Funder: The Brinson Foundation. https://brinsonfoundation.org. The funder had no role in study design, data collection and analysis, decision to publish, or preparation of the manuscript.

**Competing interests:** I have read the journal's policy and the authors of this manuscript have the following competing interests. This does not alter our adherence to PLOS ONE policies on sharing data and materials.

## Introduction

Communicating science effectively is important. Effectively mobilizing practical techniques to achieve this goal can be challenging, however. One promising science communication approach that has gained increasing attention in recent years is cinematic-style 3D data visualizations (citations anonymized for the peer review process). There is good reason to think that presenting scientific data in a cinematic manner, known as 'cinematic scientific visualization,' can bolster public engagement with scientific topics [1]. Yet, key questions remain about precisely how to use cinematic-style 3D visualization to deliver the greatest impact for the public. A key unresolved issue is the extent to which explanatory [2] or informational labels should be used in this kind of science communication. Here, we present an experimental study to address the research question: How do informational labels affect audience responses to cinematic-style scientific visualizations?

## Cinematic-style 3D data visualization

Cinematic scientific visualizations occupy a position that bridges conventional scientific visualization and artistic scientific illustration, combining the presentation of scientific data with a feature film-style artistic impression. This is a way of developing societal impacts from scientific data [3]. These visualizations are characterized by their (1) incorporation of scientific data, (2) objective to make the content comprehensible for a general audience, and (3) focus on visual attractiveness [2] (Fig 1).

As science, data management, and computer graphics evolve rapidly, the distinction between authentic scientific portrayals and their Hollywood counterparts becomes less clear. For instance, the movie "Interstellar" gained recognition for its contribution to astrophysics knowledge [4]. This merging of boundaries occurs in both directions: science enhances the realism of visual effects, while visual effects promote broader scientific accessibility [2]. By focusing on visual appeal, narrative, and filmmaking techniques, these cinematic visualizations attract millions of global viewers through entertainment and casual learning opportunities, such as museums, documentaries, YouTube, and Reddit. Simultaneously, these visualizations rely on research data and are produced in partnership with scientists. The data may be obtained from real-world sources like satellites, telescopes, microscopes, or other devices, or it might be computer-simulated based on physical laws. This unique, audience-focused scientific

**Fig 1. Hierarchy of cinematic scientific visualization needs.**

visualization approach is characterized by specific objectives, methods, and results, establishing it as a critical and emerging domain in science communication.

Traditional scientific visualization, cinematic visualizations and scientific illustration are distinct methods of science communication, each catering to distinct but overlapping audiences. While traditional visualization is predominantly geared towards scientific domain experts, the type of imagery typically seen in research publications, scientific illustration is primarily intended for a non-expert audience, featuring in both documentaries and fictional film productions. Cinematic visualization, on the other hand, attempts to bridge the gap between these two extremes. Thanks to recent breakthroughs in computer graphics technology [4, 5] and the advent of machine learning techniques [6, 7], the realm of cinematic-style visualization is becoming increasingly accessible to a wider array of scientists, thus narrowing the divide in visualizations aimed at expert and lay audiences.

Borkiewicz et al. [5] contended that scientific visualizations should utilize visual effects tools to create high-fidelity visuals that rival the quality of contemporary cinema. To achieve this objective, designers of cinematic scientific visualizations must strike a balance between precision and comprehensibility. When an image attains sufficient clarity and accuracy, attention can then be directed towards the final component, which is the aesthetic appeal. The journey from raw data to visually engaging content is laden with numerous challenges. Decisions regarding omissions, simplifications, abstractions, or the incorporation of artistic elements are generally made by data visualizers independently or in collaboration with scientists [7]. In fact, scientists may guide the process by suggesting specific narratives they want the visualization to convey, thus directing the data visualizers to emphasize these stories, potentially at the expense of other elements.

Taking an evidence-based approach means making visualization design choices not because they are the norm, or because that is what the designers personally like or prefer to implement. Instead, audience needs should be the driving force behind visualization design. However, understanding what audiences need, and using audience insights to inform practice requires research. This study is designed to deliver exactly this kind of insight to inform future science communication and science learning practice involving the use of cinematic-style 3D data visualization.

Here, we present a study evaluating the effects of public science communication techniques using research data visualizations produced by the Advanced Visualization Lab (AVL) at the National Center for Supercomputing Applications at the University of Illinois at Urbana-Champaign. The AVL works with scientists to bring their datasets to life, creating cinematic scientific visualizations for films, documentaries, and museums. The AVL's work is funded by a variety of sources, including government agencies, film companies, philanthropic foundations, and contract work. Leveraging recent technological advances, the AVL brings a cinematic level of scientific visualization to the data, creating more appealing and accessible content for the public. As an illustration of their reach, the scientific documentary "Solar Superstorms," co-produced by AVL, has been shown on international television in 16 countries, played in 86 planetariums, and has garnered 4.6 million views on YouTube alone. Additionally, this documentary is available for streaming on platforms like Amazon Prime and Magellan TV, further increasing its audience reach. The 3D data visualizations used in this documentary are the focus of the present study.

## What do we know about the impact of 3D data visualization?

Crucial discussions surrounding 3D data visualization spans an extensive array of domains, covering topics such as human perception [8, 9], data ethics and responsible representation

[10, 11], photorealism and the role of visual aesthetics in conveying information [12, 13], the application of visualization techniques in documentary filmmaking [14, 15], and the broader implications for science communication and public understanding of complex concepts [16–18]. These diverse areas of interest emphasize the interdisciplinary nature of 3D data visualization as a means of communicating science. Here, we delve deeper into the science communication and science learning implications of 3D data visualization by systematically examining effects on audiences for the first time.

Some practitioners in the 3D data visualization field refrain from employing explanatory labels in their cinematic-style visualizations, based on the unverified supposition that their inclusion may diminish the immersive quality of the visual experience for audiences. A recently published comprehensive literature review addressed this issue directly by tackling the question of whether and to what degree explanatory labels should be incorporated into 3D data visualizations aimed at public audiences [19]. This literature review deduced that the inclusion of informational labels "may come with tradeoffs between broad accessibility and precise understanding for non-technical audiences [19]. Furthermore, the article identified key variables that could potentially affect the audience's reception of cinematic scientific visualizations, including intelligibility, the content of the film, and the level of immersion. Cinematic immersion refers to the enveloping, sensory-engaging experience that allows audiences to temporarily suspend disbelief, achieving a heightened emotional and cognitive connection to the narrative. Immersion was identified as a variable worth exploring in this literature review, given it was highlighted by prior research on scientific documentary films.

Ultimately, the findings of this literature review left the question of the efficacy of labels unresolved. This is because there is no prior published peer-reviewed literature on audience responses to cinematic-style 3D data visualizations. The inconclusive nature of these results highlighted the necessity for further investigation, involving data collection with audiences, to obtain more precise and actionable insights concerning the advantages or drawbacks of integrating labels into 3D data visualizations.

## Methods

This study used a survey-based experiment to evaluate the effects of adding informational labels to space science-themed 3D data visualizations.

### Research design

To inform science communication practices, a cross-sectional online survey experiment was conducted to test the effects of informational labels on a range of audience outcomes. The survey experiment method amalgamates key aspects of observational survey methodology and experimental design (e.g., [20, 21]). Survey experiments randomly assign participants to different experimental and control conditions, and then gather data from both conditions using the same measures of attitudes, beliefs, or behaviors. Random assignment to different stimulus conditions allows for straightforward, robust causal inference in the presence of potential confounding variables. A single factorial design was used in this survey experiment.

In this case, the informational labels were designed for cinematic data visualizations featured in the film *Solar Superstorms*. This film takes viewers into the tangle of magnetic fields and superhot plasma that vent the Sun's surface energy in dramatic flares, violent solar tornadoes, and the largest eruptions in the Solar System: coronal mass ejections. The show features one of the most intensive efforts ever made to visualize the Sun's inner workings, including a series of groundbreaking scientific visualizations computed on the giant supercomputing initiative, Blue Waters, based at the National Center for Supercomputing Applications, University of Illinois.

During the pilot research to ensure the quality of the informational labels, several adjustments were made to improve clarity, relevance and impact (anonymized citation to prior work). An improved set of informational labels were used in the experimental design for this study. All aspects of the survey were in English.

## Instrument

The survey measured (a) demographic and other background profile data and (b) outcome variables encompassing different facets of audience responses to the labels. The survey assessed audience responses to the 3D data visualizations using the following sequence: (1) Participants began by completing the consent and demographic sections of the survey, (2) Participants were randomly assigned to see one of two versions of an approximately 10-minute video containing a truncated version of the Solar Superstorms narrated documentary film that showed only the cinematic data visualizations (and cut out other aspects of the film)–Version A contained explanatory labels and Version B did not, (3) Participants were asked to provide feedback about the overall film experience, (4) finally, they answered a series of attitudinal and learning-related questions to measure their level of understanding of key content covered in the film.

The survey included the following question and response types:

- Participants were asked to rate the overall quality of the film clip that they viewed on a scale of 0–100.

- Semantic differentials were used to gauge audience reactions by evaluating their responses between two contrasting adjectives, for instance, confusing / clear, interesting / uninteresting, and informative / uninformative.

**Immersion and perceived scientific realism.** Level of agreement scales were used to measure immersion and perceived scientific realism, on a 7-point Likert-type scale. The following statements evaluated whether labels affected how immersive the viewing experience felt for audiences:

- 'It was no effort to keep my mind on what was happening during the video clip'

- 'My attention was focused entirely on the video clip'

- 'I felt as if I were in space, instead of watching the video clip on a screen'

The survey also probed participants' assessments of the film to see if they understood that these were data visualizations being displayed, or whether they assumed these were scientific illustrations. This variable was assessed using a set of three level of agreement scale statements:

- 'The video clip is based on real scientific data.'

- 'The video clip is showing real images seen through a telescope.'

- 'The video clip shows an artist's illustration of what they think The Sun looks like.'

The survey was conducted via the Qualia Analytics platform.

## Information labels introduced for the experimental condition

The visualizations created for the *Solar Superstorms* film, originally released for dome shows, did not include informational labels. The reason labels were avoided was a concern that they would feel intrusive and detract from the immersive experience of the film. To understand if

this concern is valid, a set of informational labels was created for visualizations taken from the documentary and presented to the test audience in a new, shortened version of the film that only contained the 3D data visualizations (in their original sequence, but without other kinds of content intermingled). This new film clip was available in two forms: One with the labels (experimental/treatment condition) and one using the original versions of the visualizations without the labels (control condition). This means the independent variable for this study was the presence/absence of informational labels alongside the cinematic-style data visualizations.

Informational labels used in the experimental condition for this survey are displayed below in Figs 2A–2C, 3A–3D, 4A, 4C, 5A, 5B, and 6A–6C with additional explanation in each caption. In Fig 4B, an attempt at expressing scale was redesigned after feedback during the pilot phase and replaced with Fig 4C. All labels were absent in the control condition.

The dependent variables used to operationalize audience responses to the cinematic-style 3D data visualizations are as follows: Ratings of quality on a 0–100 scale, emotional responses using semantic differentials (opposing adjective pairs), immersion and perceived scientific realism using level of agreement items.

## Sampling

The aim for sampling here was to gather participants who would be open to viewing a scientific documentary film, the primary target audience for the kind of cinematic-style data visualizations this study addresses. Representative sampling is less important in survey experiments of this kind due to homogeneity of treatment effects. However, we have documented our approach to gather respondents below. The stated achieved sample for this survey experiment is 577 completed responses. A survey responses were retained if at least one outcome variable was completed in full: Most survey questions were not mandatory-response items.

Ultimately, 48% (n = 276) of respondents viewed the 'labels' experimental condition of the video, while 52% (n = 298) so the 'no labels' control version. Posthoc power analyses for each of the variables confirmed that the experiment was well-powered:

A post-hoc power analysis for the t-test comparing two independent means was conducted. The effect size was set to 0.5 with a two-tailed significance level ($\alpha$) of 0.05 for this analysis: Group 1 = 276; Group 2 = 298. Results showed a critical t-value of 1.9642 (df = 572). The analysis showed a high power of 0.9999703, indicated a very low risk of Type II error. The noncentrality parameter was 5.9852.

A pool of participants for this survey was recruited through two main routes. The main sampling approach was to post a participant recruitment message on Facebook, using a $300 advertising boost. The advertisement noted who was carrying out the survey and why (to help improve documentary films and science communication), summarized what would be asked of participants (to watch a 9-minute video and respond to a 10-minute survey), and offered a chance to win a $50 gift card (Fig 7).

This recruitment route was aimed at recruiting general public audience members with experience or interest in scientific documentary films or related video content. This approach yielded the vast majority of the respondents for this study (n = 497 fully completed and 258 partially completed responses).

The other major recruitment method for attracting general public audience members to the study was to send a follow-up request to survey respondents who previously completed a public consultation survey about the future of space science. This follow-up invitation was restricted to those who had explicitly indicated interest in receiving additional invitations to participate in social research relating to space science. This route delivered a minority of the study respondents (n = 65 fully completed and 4 partially completed responses).

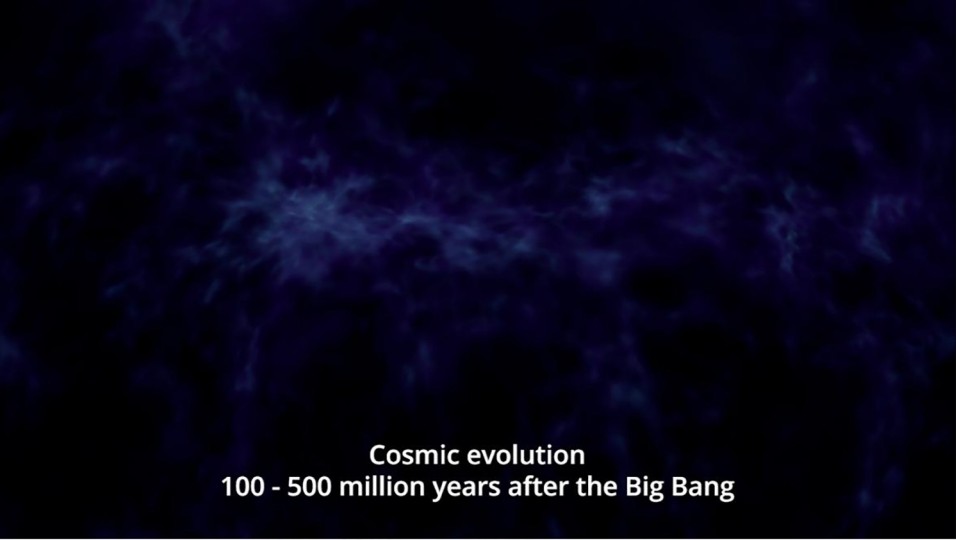

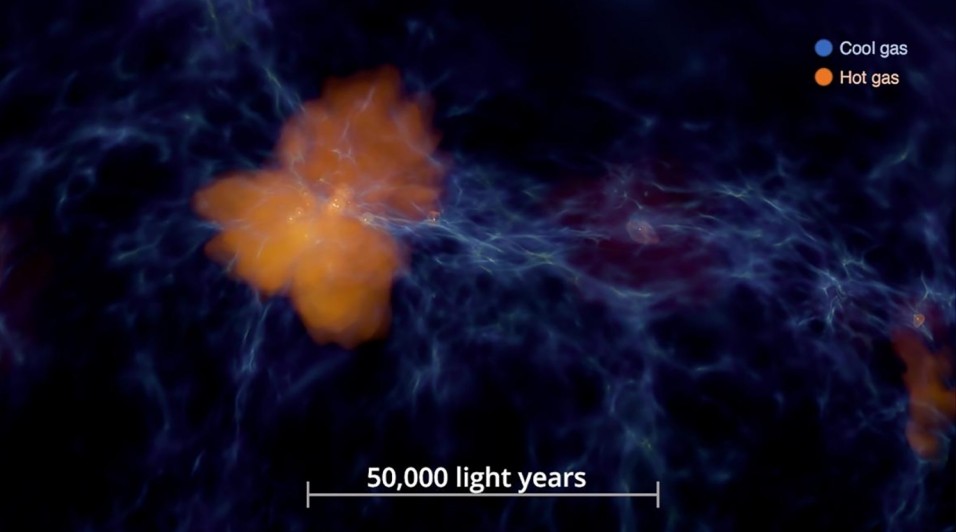

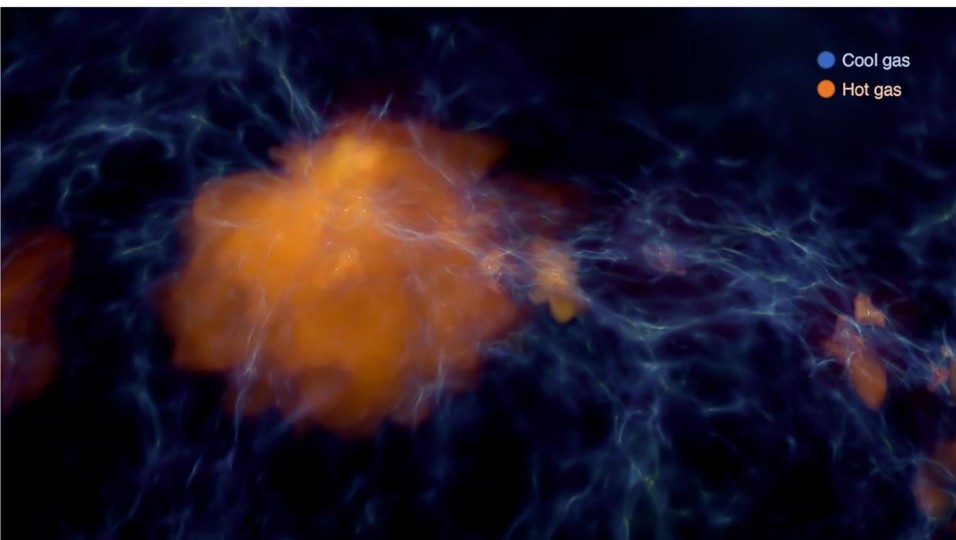

**Fig 2. a. Cosmic evolution 100–500 million years after the Big Bang.** Label adds context regarding formation of the early universe, when the first stars were forming. **b. 50,000 light years.** A scale bar, as is familiar from common road maps, here suggests the immense size of the features displayed. **c. Cool Gas, Hot Gas.** As the narration explains the gas is color-coded according to its temperature, these labels scaffold audience understanding of the visualized data.

Attempts to circulate the survey request through existing email lists for organizations with relevant interests such Science News and Magellan TV were largely unsuccessful, resulting in a grand total of just 15 fully completed responses for this study (and 0 partial completions).

## Participants

The achieved sample had a mean age of 41.9. The age distribution of the participants was as follows: 2% (n = 8) were 19 years old or younger, 11% (n = 55) between 20–24 years, 15% (n = 72) between 25–29 years, 13% (n = 65) between 30–34 years, 10% (n = 49) between 35–39 years, 9% (n = 45) between 40–44 years, 5% (n = 25) between 45–49 years, 10% (n = 48) between 50–54 years, 6% (n = 31) between 55–59 years, 8% (n = 37) between 60–64 years, 7% (n = 36) between 65–69 years, 2% (n = 12) between 70–74 years, 1% (n = 4) between 75–79 years, and less than 1% (n = 2) between 80–84 years.

The gender distribution of the sample skewed towards men, with 30% (n = 144) identifying as female, while 70% (n = 337) identified as male. No respondents self-identified as non-binary.

The sample included a disproportionately high number of participants with postgraduate degrees. The participants' highest educational qualifications were classified into four categories: no educational qualification, non-degree educational qualification, university degree (e.g., Bachelor of Arts or Bachelor of Science), and postgraduate degree (e.g., Master's/PhD degrees). The breakdown of the participants' highest qualifications is as follows: 2% (n = 10) had no educational qualification, 22% (n = 107) held a non-degree educational qualification, 35% (n = 168) had earned a university degree, and notably, 40% (n = 191) possessed a postgraduate degree.

Among those who had degrees, the subject areas of the respondents' most recent degrees were requested to gain a better understanding of their academic backgrounds. The breakdown of the respondents' most recent degrees is as follows: 2% (n = 5) in Science Communication, 4% (n = 13) in Mathematics, 7% (n = 22) in Humanities, 10% (n = 32) in Social Sciences, 20% (n = 63) in Technology, 27% (n = 84) in Engineering, and 29% (n = 91) in Science.

Respondents' country of residence was the United States for 33% (n = 164) of respondents. Those residing outside the United States (67%, n = 327) were distributed across 37 countries. The distribution of the participants by country was as follows (from least to most prevalent): Japan, Kenya, Algeria, United Arab Emirates, Argentina, Iran, Philippines, Norway, Slovakia, Réunion, China, Anguilla, Ireland, Czechia, and Nicaragua each had 1 respondent; Denmark, Sweden, Austria, Russia, Mexico, and Estonia each had 2 respondents; Finland, Hungary, and Poland each had 3 respondents; Australia, Canada, and Romania each had 4 respondents; Switzerland had 5 respondents; Greece had 6 respondents; Portugal and Belgium each had 7 respondents; the United Kingdom had 8 respondents; Spain had 18 respondents (6%); the Netherlands had 21 respondents (6%); Germany had 54 respondents (17%); Italy had 62 respondents (19%); and France had the highest number of respondents with 91 (28%). The remaining respondents did not indicate their country of residence.

The breakdown of the participants by race and ethnicity, among those who responded to this question that appeared for US-based respondents only, was distributed as follows: Native Hawaiian or Pacific Islander represented 1% (n = 2) of the sample, American Indian or Alaska

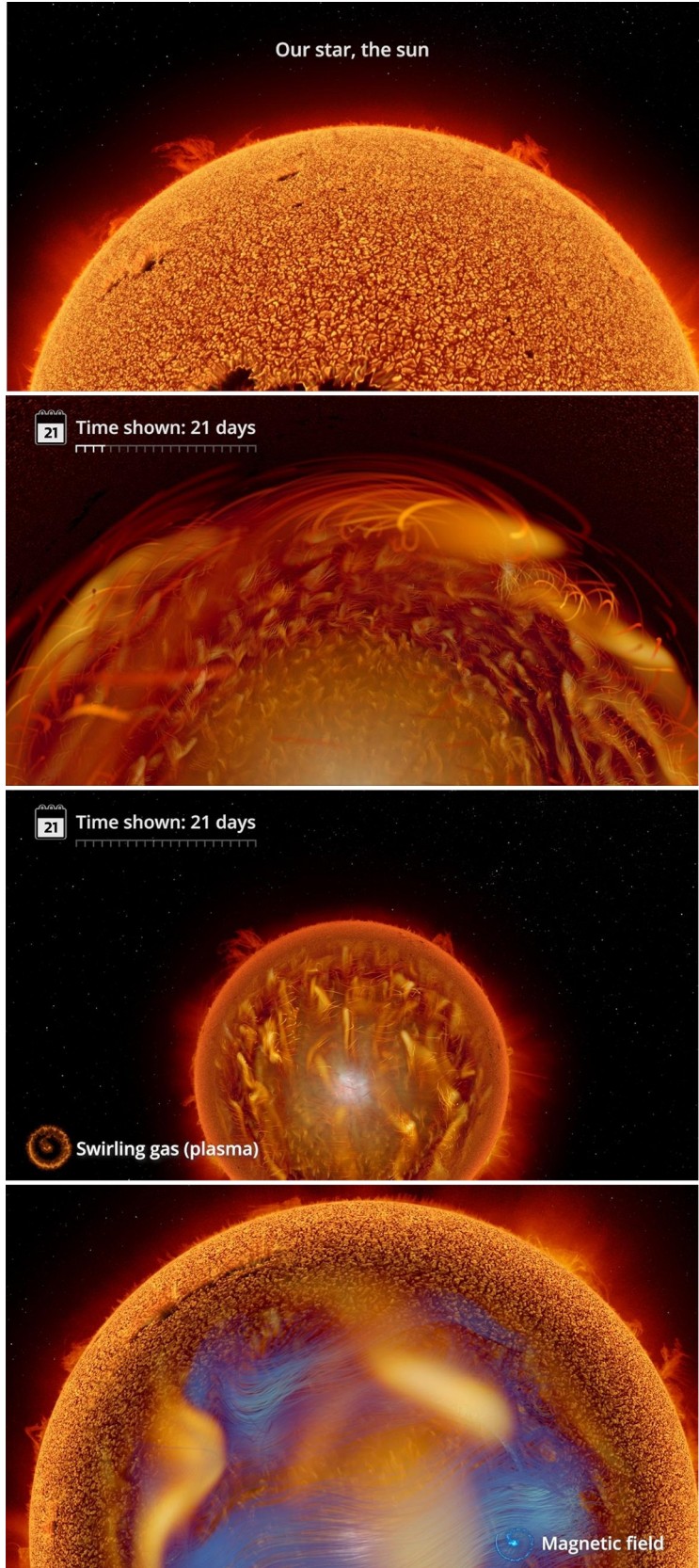

**Fig 3. a. Our Star, the Sun.** This simple label reinforces the fact that our sun is a star and not some other special kind of celestial body. **b. Time shown: 21 days.** The initial version of the label (not shown) was labeled with just the static calendar page. Here, an animated time bar was added to show time progressing dynamically. **c. Swirling gas (plasma).** An animated swirl (bottom left) was added, matched to the color of the plasma in the visualization, to clarify which data was being displayed. **d. Magnetic field.** To introduce a new element–the magnetic field, shown in blue–we added a new swirl icon of that color.

Native made up 4% (n = 6), Hispanic, Latino, or Spanish constituted 4% (n = 7), Black or African American accounted for 10% (n = 17), Asian comprised 14% (n = 23), and White, being the largest group, represented 75% (n = 122).

## Data analysis

Percentages were rounded to the nearest integer. Statistically significant results are reported at $\alpha < 0.05$ throughout this article. Independent samples t-tests were used for dependent variables comprised of continuous data. Independent samples t-tests operate on the assumption of homogeneity of variances. To confirm this, the Levene's test statistic was used to ensure that the equality of variances assumption of the statistical test have not been violated.

**Ethical clearance.** Institutional Review Board approval for this research was granted by the University of Illinois at Urbana-Champaign on May 31[st], 2022 (protocol number 23080) prior to the commencement of any contact with potential research participants. Data collection took place from January to April 2023.

## Results

Here, we present the empirical results from the survey experiment, focusing on the comparison between audience outcomes for the experimental (labels) and control (no labels) versions of the video that was shown to respondents. These results clarify whether the use of informational labels in cinematic-style scientific visualization affects audience outcomes, including ratings of quality, emotional responses, immersion and perceived scientific realism.

### Video quality rating

Audience assessments skewed positive, with a mean score of 80.74 out of 100 overall (n = 574). The experimental (labels) condition had a higher proportion of respondents giving the highest level of video quality (i.e., in the 90–100 range), with 28% at this level (compared to 21% for the control). Indeed, an independent samples t-test revealed a statistically significant difference in video quality ratings between the experimental and control conditions ($t(572) = -2.801$, $p = .005$). Specifically, the experimental condition gained a mean quality rating of 82.70 (SD = 14.033, n = 276), while the control condition mean was 78.92 (SD = 17.845, n = 298).

A non-significant Levene's test result ($F = 3.627$, $p = .057$) confirmed equality of variances (i.e., equal variances assumed for the t-test), upholding this key parametric assumption for the t-test.

The effect size of the difference between the two groups was quantified using Cohen's *d*. Cohen's *d* tells us how large the difference is between the two groups being compared, relative to the variability within the groups. Here, the point estimate for Cohen's *d* was -0.234, indicating a small to moderate effect size.

### Assessments of the 3D visualization film

Respondents were asked to assess the documentary film clip they watched, which featured 3D scientific visualizations. Survey results were analyzed to assess participants' perceptions with

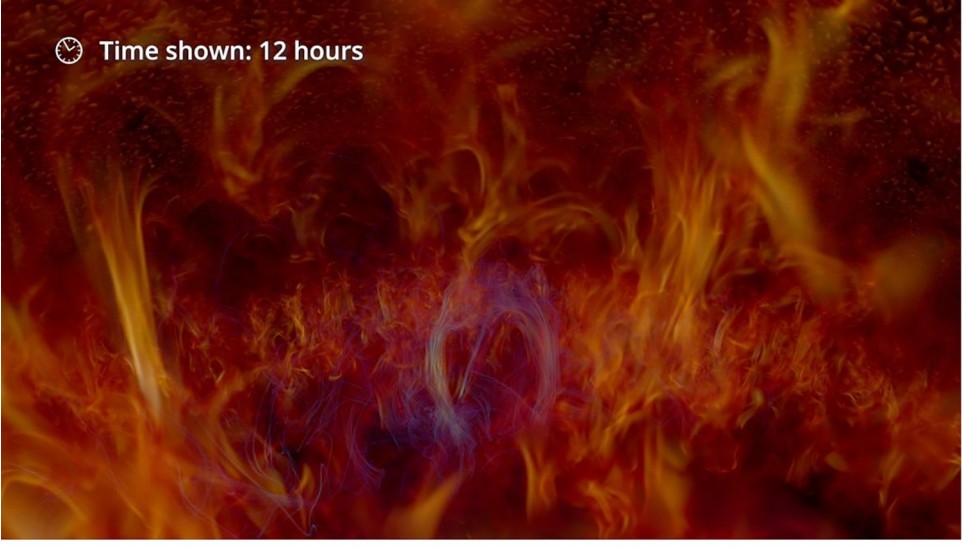

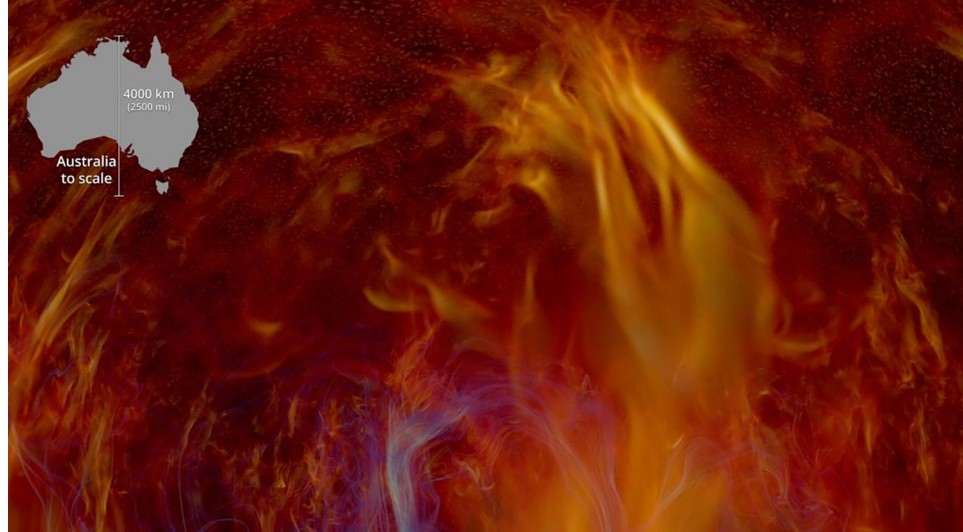

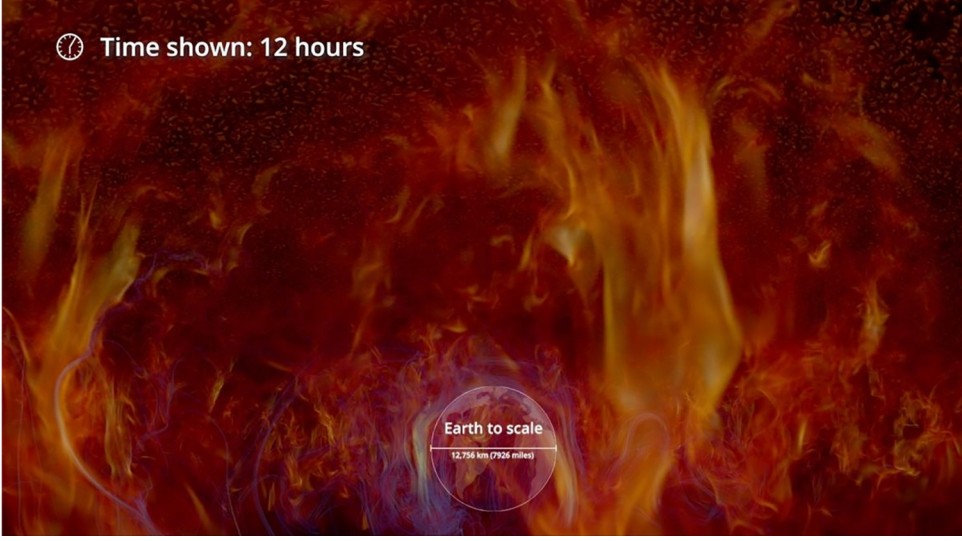

**Fig 4. a. Time shown: 12 hours.** A view of a smaller portion of the sun, with reduced scales in both space and time. To show the passage of time in hours, an animated clock icon was added. **b. Spatial scale: Australia, lacking clear 3D context.** A first attempt to communicate the spatial scale used a picture of Australia as a foreground element overlaid on the image. But several respondents during the pilot phase found this label confusing. To which part of the 3D scene was it to be compared? **c. Redesigned spatial scale: Earth, aligned with a discrete 3D feature.** The AVL team adapted our labels based on feedback. Here, a compact feature within the 3D scene, a magnetic loop, has a diameter comparable to that of the Earth. Juxtaposing them provides a depth cue for the scale element, and aims to improve the clarity and intelligibility of the visualization.

and without labels. Participants rated the videos along various semantic differentials, including interest, fascination, excitement, stimulation, clarity, realism, scientific nature, aesthetic appeal, informativeness, and level of alarm activated by the video (which focused on the theme of solar superstorms).

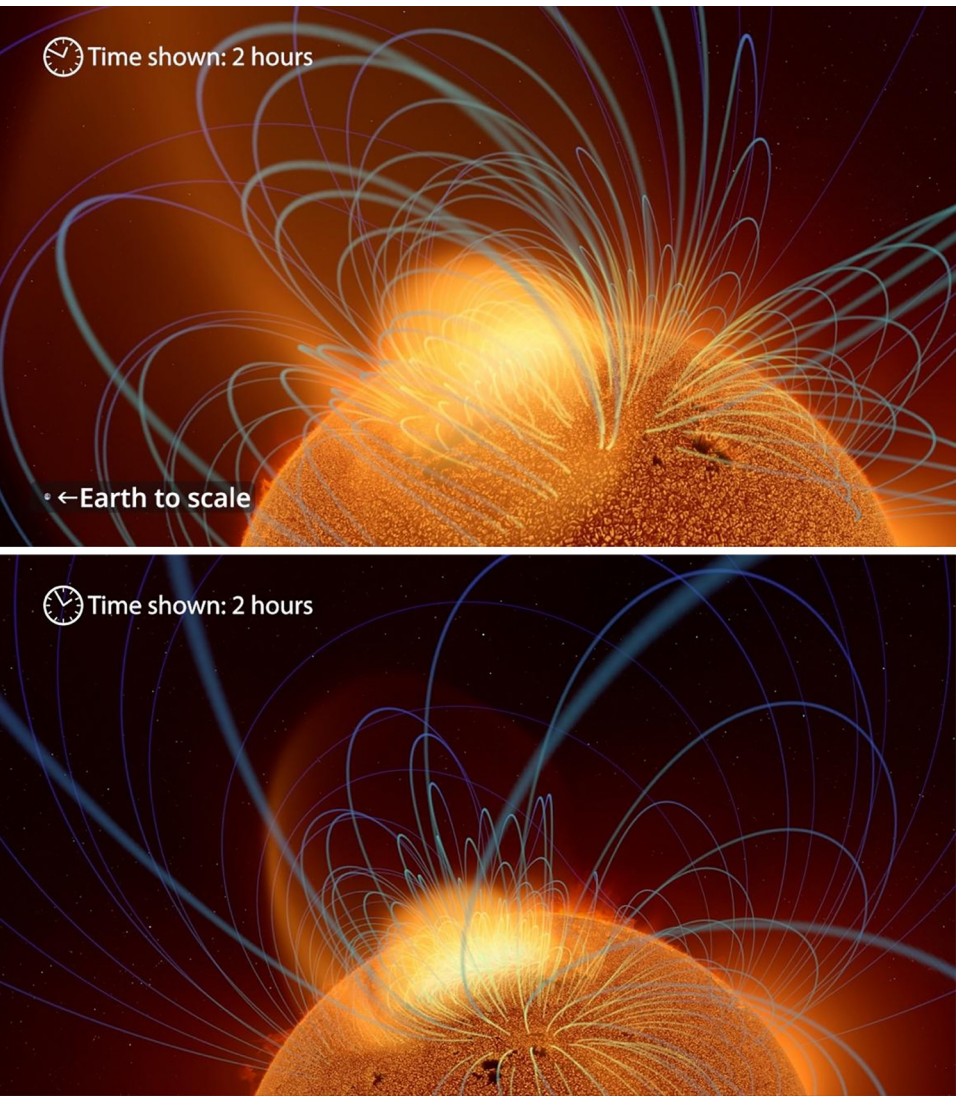

**Fig 5. a. Solar exterior: time and space scales.** A view of the exterior of the sun, again showing Earth to scale and giving the audience a sense of the timescale. Time shown: 2 hours (top, left), and Earth to scale (bottom, right). **b. Passage of time.** The clock moves forward as the visualization progresses, continuing to represent the timescale.

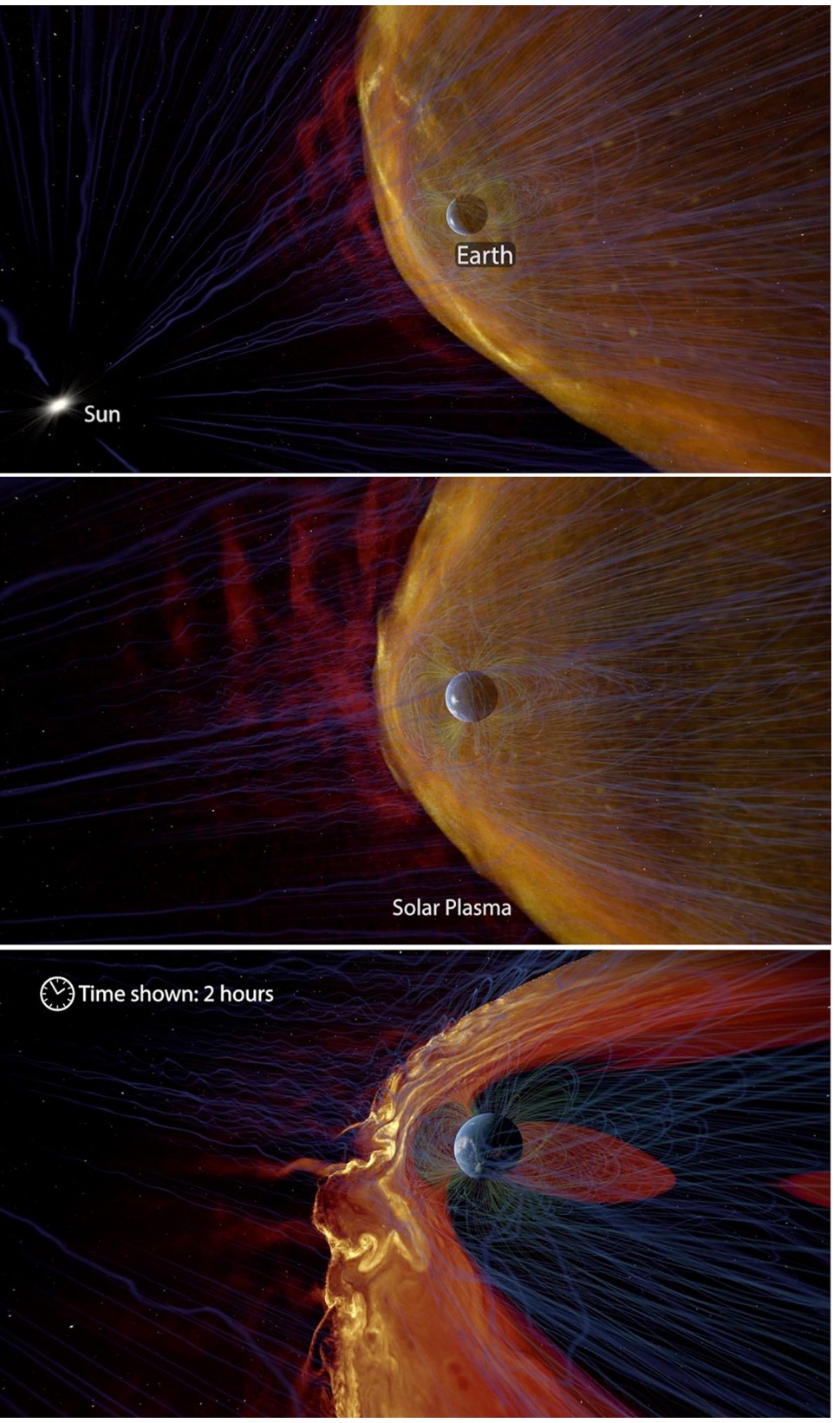

**Fig 6. a. Sun, Earth.** Simple text labels mark the location of the sun, and clarify that we are looking at Earth rather than another planet. **b. Solar Plasma.** In addition to the label text, a visual effect highlights the solar plasma as the narrator mentions it. **c. Time shown: 2 hours.** As before, a moving clock marks the passage of time.

While there were some differences between experimental and control groups evident in the descriptive statistics, none of these rose to the level of statistical significance (Table 1). Overall, these results suggest that the presence of labels in the videos did not affect participants' perceptions of the video across the dimensions assessed. That is, there was a similar assessment of the video across both conditions. Indeed, the overall distribution of participants' ratings skewed positive across almost all the categories. For example, on the interesting-uninteresting scale the results for experimental and control conditions combined were as follows: 2% (n = 10) rated the film clips as -3 (Uninteresting), 4% (n = 23) rated them as -2, 6% (n = 33) rated them as -1, 5% (n = 28) had a neutral perception, 12% (n = 69) rated the film clips as +1, 30% (n = 167) rated them as +2, and the largest group, 42% (n = 236), rated the film clips as +3 (Interesting). Overall, these results demonstrate that most participants found the videos to be interesting, with 72% rating the videos as either +2 or +3.

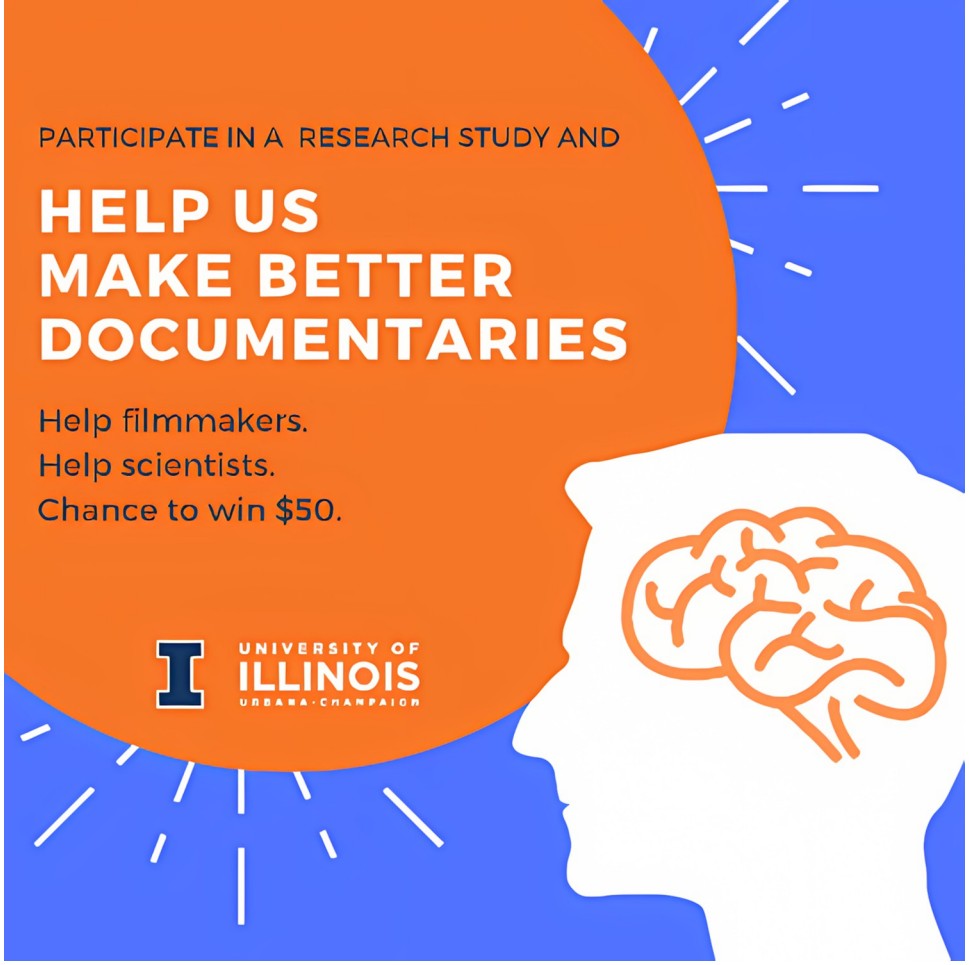

**Fig 7. Image posted on social media accounts for respondent recruitment.**

**Table 1. Assessments of video by audiences.**

| Semantic differential | Video Displayed | t | p | df | N | Mean | SD |
|---|---|---|---|---|---|---|---|
| Uninteresting–Interesting | No Labels | -1.803 | .072 | 564 | 294 | 1.66 | 1.644 |
| | Labels | | | | 272 | 1.89 | 1.362 |
| Boring—Fascinating | No Labels | -1.202 | .230 | 566 | 296 | 1.45 | 1.636 |
| | Labels | | | | 272 | 1.60 | 1.462 |
| Mundane—Exciting | No Labels | .864 | .388 | 564 | 295 | 1.22 | 1.456 |
| | Labels | | | | 271 | 1.12 | 1.451 |
| Dull—Stimulating | No Labels | -1.090 | .276 | 566 | 295 | 1.27 | 1.549 |
| | Labels | | | | 273 | 1.40 | 1.393 |
| Confusing—Clear | No Labels | -1.351 | .177 | 567 | 296 | 1.47 | 1.545 |
| | Labels | | | | 273 | 1.64 | 1.544 |
| Unrealistic—Realistic | No Labels | -1.580 | .115 | 565 | 295 | 1.22 | 1.537 |
| | Labels | | | | 272 | 1.42 | 1.448 |
| Unscientific—Scientific | No Labels | -.331 | .741 | 566 | 295 | 1.87 | 1.466 |
| | Labels | | | | 273 | 1.91 | 1.461 |
| Ugly–Beautiful | No Labels | -1.597 | .111 | 564 | 294 | 1.54 | 1.643 |
| | Labels | | | | 272 | 1.75 | 1.459 |
| Uninformative—Informative | No Labels | -1.797 | .073 | 566 | 296 | 1.80 | 1.514 |
| | Labels | | | | 272 | 2.01 | 1.356 |
| Alarming—Reassuring | No Labels | -.551 | .582 | 566 | 295 | .16 | 1.462 |
| | Labels | | | | 273 | .22 | 1.457 |

NOTE: No statistically significant differences were identified when comparing responses to the No Labels and Labels versions of the video using an independent samples *t*-test. Participants rated the videos using a series of 7-point semantic differential scale, ranging from -3 (negative part of the scale) to +3 (positive part of the scale), with 0 as a neutral mid-point.

This generally positive response to the video, regardless of experimental condition, was relatively consistent. The only exception was the scale 'alarming–reassuring', which garnered a mean score close to neutral.

## Immersive viewing experience

There were no statistically significant effects found for the three items measuring immersion (Table 2), that is, the results were similar for both the labels and no labels versions of the film clip.

## Scientific realism of the 3D data visualizations

There were no statistically significant differences between treatment and control groups for any of the items measuring perceived scientific realism (Table 3).

**Table 2. Immersive viewing experience self-assessment.**

| Question | Displayed | t | df | p | N | Mean | SD |
|---|---|---|---|---|---|---|---|
| Followed video effortlessly | No Labels | -.842 | 562 | .400 | 295 | 5.12 | 1.774 |
| | Labels | | | | 269 | 5.25 | 1.695 |
| Attention focused on video | No Labels | -.372 | 564 | .710 | 298 | 5.46 | 1.414 |
| | Labels | | | | 268 | 5.50 | 1.394 |
| Felt like viewing in space | No Labels | 1.078 | 558 | .282 | 293 | 4.21 | 1.868 |
| | Labels | | | | 267 | 4.04 | 1.783 |

NOTE: No statistically significant differences were identified when comparing responses to the No Labels and Labels versions of the film clip using an independent samples *t*-test. These measures are on 7-point scales, ranging from strongly agree (7/7) to strongly disagree (1/7), with a 'neutral' mid-point (4/7).

**Table 3. Audience assessment of scientific realism of video.**

| Survey question | Video | t | p | df | N | Mean | SD |
|---|---|---|---|---|---|---|---|
| Video based on scientific data | No Labels | -1.214 | .225 | 557 | 293 | 5.95 | 1.171 |
| | Labels | | | | 266 | 6.06 | 1.060 |
| Video used real images | No Labels | .585 | .559 | 557 | 293 | 2.72 | 2.048 |
| | Labels | | | | 266 | 2.62 | 1.966 |
| Sun was an illustration | No Labels | .175 | .861 | 555 | 295 | 4.97 | 1.814 |
| | Labels | | | | 262 | 4.94 | 1.837 |

NOTE: No statistically significant differences were identified when comparing responses to the No Labels and Labels versions of the video using an independent samples *t*-test. These measures are on 7-point scales, ranging from strongly agree (7/7) to strongly disagree (1/7), with a 'neutral' mid-point (4/7).

Results showed that participants in both experimental and control conditions were clear that the video they watched (comprised of 3D data visualizations) was created using scientific data ('The video clip is based on real scientific data'). 38% (n = 217) of all respondents *strongly agreed* with this statement, followed closely by 40% (n = 229) who *agreed*. Additionally, 12% (n = 66) *somewhat agreed*, indicating that a combined 90% of participants recognized the video's basis in authentic scientific information. In contrast, 7% (n = 40) of the respondents remained *neutral*, while only a small minority expressed disagreement. Specifically, 1% (n = 8) *somewhat disagreed*, another 1% (n = 7) *disagreed*, and a mere 1% (n = 4) *strongly disagreed* with the statement that the video clip was based on real scientific data. The median, mode, and mean responses were all *agree* (6 out of 7), with a standard deviation of 1.13 and a variance of 1.28. This lack of variability in responses left little room for possible differences between experimental and control groups on this measure.

To further investigate the precision of audiences' understanding that the video they saw was comprised of 3D data visualizations, we also assessed the level of agreement with the statement: 'The video clip is showing real images seen through a telescope.'

Across both experimental and control groups, most participants expressed disagreement with this reverse-coded item, thereby signaling their accurate understanding of the status of the video. Specifically, 4% (n = 23) of all respondents somewhat disagreed, 16% (n = 94) disagreed, and a substantial 46% (n = 262) strongly disagreed with the notion that the video clip showed real images seen through a telescope. A small percentage of respondents *strongly agreed* (5%, n = 27) or *agreed* (10%, n = 57) with the statement. A slightly higher percentage *somewhat agreed* (11%, n = 63). Meanwhile, 8% (n = 45) of the respondents remained *neutral*. For this item, the median was *disagree* (2 out of 7), the mode was *strongly disagree* (1/7) and the mean was *somewhat disagree* (2.71), with a standard deviation of 2.02 and variance of 4.1. These results indicate that there was a high level of clarity among audiences about the fact that the video contained visuals that were based on real data but were not directly photographed or recorded. This means they likely understood the video contained data visualizations, although there was more variation in the responses to this question compared to the previous one about the video being based on real scientific data.

Participants were also asked to express their level of agreement with the statement, 'The video clip shows an artist's illustration of what they think The Sun looks like,' to finalize the evaluation of whether they understood they had seen data visualizations. The responses showed that 19% (n = 110) of all respondents to this reverse-coded item strongly agreed, and 31% (n = 179) agreed with the statement. Furthermore, 18% (n = 105) somewhat agreed, indicating that a combined 68% of participants believed that the Sun was an illustration. In contrast, 8% (n = 48) of the respondents remained neutral, while a minority of participants

disagreed with the statement. Specifically, 6% (n = 33) *somewhat disagreed*, 10% (n = 58) *disagreed*, and 6% (n = 37) strongly disagreed with the notion that the Sun was an illustration. The mean for all respondents to this question was 4.94 out of 7 (*somewhat agree*), suggesting a less confident view compared to the answers regarding the video being based on real scientific data. Median and mode were still *agree* (6 out of 7), with a standard deviation of 1.83 and variance of 3.36.

## Discussion

This survey experiment explored differences in audience outcomes by comparing an experimental group (labels) and a control group (no labels) that watched the same cinematic-style scientific visualization documentary film clip. A mean score of 80.74 out of 100 represented a generally positive sentiment about video quality (n = 574). A statistically significant difference between the two groups emerged, with a mean score of 82.70 for the experimental group compared to 78.92 for the control group. This suggests that labels somewhat influenced audience perceptions of film quality. This means that well-designed informational labels can improve the audience experience.

The experiment's detailed assessment evaluating audience responses across several semantic differentials, such as uninteresting-interesting and ugly-beautifu*l* revealed no statistically significant differences between conditions. Potential differences in immersive viewing experience and scientific realism were also investigated. The three items used to evaluate the immersive nature of the videos–ease of mind, attention focus, and feeling as if in space–showed no statistically significant differences between the labeled and unlabeled videos. This shows that informational labels had no material effect on how immersive the viewing experience was for audiences. The 3D visuals were the constant between the experimental and control groups, so this finding suggests that the story structure, voiceover narration and audio-visual content are driving the impact of the 3D visualization on experiential immersion indicators, in this case, ease of mind, attention focus, and feeling as if in space.

Equally, when assessing the scientific realism of the 3D data visualizations, there were no significant differences between the experimental and control groups. This suggests that adding informational labels did not compromise the quality of the viewing experience. This contrasts with concerns that such informational elements could undermine audience immersion and enjoyment. At the time, it again indicates that it is the story, narrative and audio-visual aspects of the experience that are paramount in driving audience perceptions of scientific realism.

3D data visualization is becoming crucial for science communication, particularly for engaging public audiences through documentaries and streaming services. This paper presents a research study designed to reveal effective techniques in 3D visualization of scientific data for public audiences. This audience research project was designed to shape design decision-making to increase the impact of cinematic-style data visualizations. Cinematic scientific visualization uses filmmaking techniques to present complex scientific datasets in a visually compelling way while prioritizing audience understanding. It currently occupies a sliver of space along the spectrum between "traditional data visualization" and "scientific illustration". However, its broad appeal and ever-expanding technical feasibility will see it continue to grow as a field and gain ground in the overlapping domains of education and entertainment. The question is how this form of science communication can be more effective. To do this, we believe in taking an evidence-based communication approach, with audience research to systematically test different design options.

By using cinematic-style 3D data visualization, researchers and science communicators can provide a new perspective on scientific research and discoveries. This approach allows viewers

to see the subject matter in a way that might not be possible with traditional forms of communication, such as text or 2D images. Moreover, 3D data visualization has the potential to capture viewers' attention and create a lasting impression. The use of motion, color, and other visual elements can make the information more memorable and engaging.

Jensen & Gerber [22] champion evidence-based science communication, urging adaptation to diverse audiences and updating practices using audience research [23]. Despite the growing popularity of evidence-based science communication as a goal, concrete published examples clarify what this looks like in practice remain rare [24]. Indeed, a clear roadmap is glaringly absent in science communication and 3D data visualization literature, leaving practitioners to grapple with key decisions.

Unleashing the full power of evidence in science communication is a fundamental step towards enhancing the effectiveness of public engagement with science. When venturing into uncharted territories with cinematic-style scientific visualizations, exclusive reliance on professional intuition can fall short. In these cases, empirical evidence can serve as a valuable compass in navigating communication decision-making. Audience research can give practitioners valuable insights into what works (or fails) when conveying complex scientific concepts. As we continue to explore and refine this approach, we can enhance our understanding of the world around us and make scientific information more accessible to a wider audience.

## Acknowledgments

We would like to thank the research participants. User experience improvements to the survey instrument were made by Dr. Aaron M. Jensen with support from Juan Barbosa at the Institute for Methods Innovation, constituting a non-financial contribution to the project.

## Author Contributions

**Conceptualization:** Eric Allen Jensen, Kalina Borkiewicz, Jill P. Naiman, Stuart Levy, Jeff Carpenter.

**Data curation:** Eric Allen Jensen.

**Formal analysis:** Eric Allen Jensen.

**Funding acquisition:** Kalina Borkiewicz.

**Investigation:** Eric Allen Jensen, Kalina Borkiewicz, Stuart Levy, Jeff Carpenter.

**Methodology:** Eric Allen Jensen, Kalina Borkiewicz, Stuart Levy, Jeff Carpenter.

**Project administration:** Eric Allen Jensen, Kalina Borkiewicz, Stuart Levy, Jeff Carpenter.

**Resources:** Eric Allen Jensen, Kalina Borkiewicz, Stuart Levy, Jeff Carpenter.

**Software:** Eric Allen Jensen, Kalina Borkiewicz, Stuart Levy, Jeff Carpenter.

**Supervision:** Eric Allen Jensen, Kalina Borkiewicz.

**Validation:** Eric Allen Jensen, Stuart Levy.

**Visualization:** Eric Allen Jensen, Stuart Levy, Jeff Carpenter.

**Writing – original draft:** Eric Allen Jensen.

**Writing – review & editing:** Eric Allen Jensen, Kalina Borkiewicz, Jill P. Naiman, Stuart Levy, Jeff Carpenter.

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
