## [Decision Letter · Decision Letter 0]

20 Sep 2023

PONE-D-23-23886Picture Perfect Science Communication: Informational Labels in 3D Data Visualization for Public EngagementPLOS ONE

Dear Dr. Jensen,

Thank you for submitting your manuscript to PLOS ONE. After careful consideration, we feel that it has merit but does not fully meet PLOS ONE’s publication criteria as it currently stands. Therefore, we invite you to submit a revised version of the manuscript that addresses the points raised during the review process.

 Though the topic is interesting, the aim and motivation for the study (including the formal proposition of research questions or hypotheses to be tested) and the methodology (which may look too simple to warrant significant results) are lacking. You should refer to the specific comments made by the reviewer to improve your paper. 

We look forward to receiving your revised manuscript.

Kind regards,

Maurizio Naldi

Academic Editor

PLOS ONE

Journal Requirements:

"Would like to thank the research participants. This research was funded by The Brinson Foundation as part of the Civic Science Fellows program. The SciWise initiative (sciwise.org) also provided support via the survey instrument. User experience improvements to the survey instrument were made by Dr. Aaron M. Jensen (Institute for Methods Innovation, methodsinnovation.org). "

"All authors worked on the grant, but PI was KB. There was no grant number. Funder: The Brinson Foundation. https://brinsonfoundation.org

"I have read the journal's policy and the authors of this manuscript have the following competing interests."

6. We note that Figure 1 in your submission contain copyrighted images. All PLOS content is published under the Creative Commons Attribution License (CC BY 4.0), which means that the manuscript, images, and Supporting Information files will be freely available online, and any third party is permitted to access, download, copy, distribute, and use these materials in any way, even commercially, with proper attribution. For more information, see our copyright guidelines: http://journals.plos.org/plosone/s/licenses-and-copyright.

Reviewers' comments:

Reviewer's Responses to Questions

**Comments to the Author**

1. Is the manuscript technically sound, and do the data support the conclusions?

Reviewer #1: Partly

2. Has the statistical analysis been performed appropriately and rigorously? 

Reviewer #1: Yes

3. Have the authors made all data underlying the findings in their manuscript fully available?

Reviewer #1: Yes

4. Is the manuscript presented in an intelligible fashion and written in standard English?

Reviewer #1: Yes

5. Review Comments to the Author

Reviewer #1: For starters, this paper has an interesting topic that has the potential to advance the understanding of science communication through data visualization. That being said, this paper requires a lot of improvement before it is ready for publication.

There are some fundamental aspects of a typical research article (i.e., a theoretical framework, formally proposed hypotheses and research questions, etc.) that are lacking in this paper. I have listed some of the main things that can be addressed. This is not an exhausted list.

--

Major issues

1. While studies don't need to be complicated to be valuable, your study seems too simple (i.e. this is a single factorial design with two conditions). Many studies have examined data visualization in science communication. I would suggest one or more of the following to fix this:

a. add in another variable,

b. provide further reasoning for your paper's ability to move science communication in data visualization literature forward, or

c. mention your study's simplicity as a limitation in your limitations section

*Also, you could argue that your are adding to existing, similar lit related to data visualization in science communication as a form of replication to enhance this research area in science communication.

2. While you mention what you are doing, there are no formally proposed hypotheses or research questions in this paper. These are important to have, because they provide structure for the rest of your paper.

Your lack of research questions and hypotheses make your results section confusing in terms of what you are trying to specifically accomplish with your collected data in this paper (aside from your overarching question on your audience’s perceptions of videos with or without labels).

Hypotheses can straighten out your paper for you and the reader pertaining to what variables you are measuring, hint at what statistical analyses you will be using, and whether your findings were significant or not. It makes writing your discussion a lot easier too.

3. On pg. 6, in the last paragraph of your literature review you mentioned entities like level of immersion yet there's no section in your literature review that covers this topic. Level of immersion is an important concept to address and elaborate on for papers covering anything with visual technology or media.

4. There is no formally introduced theoretical framework in this paper. Is it the hierarchy of cinematic scientific visualization needs? If so, give it its own section and elaborate on how this theory pertains to your study in your lit review. Otherwise, there are many theoretical frameworks (social and psychological) that would work well with this article’s topic.

5. At the beginning of the method section, you mentioned that this is an experiment, but then you don't include some specificities involved in an experimental design. For example, your factorial design is not listed (it would be a single factorial design with 2 conditions for this study).

You mentioned conditions, but you are not concrete enough about what independent variables are being manipulated within each condition. Explaining these things and including a diagram of the variables you're manipulating, your mediating and moderating variables, and your outcome variables would be really helpful to those who are and are not familiar with your research.

6. Your pilot study explanation is not clear enough. You need to explain what these adjustments were to your labels. You should add-in statistics substantiating the validity of your examined variables in your pilot to enhance the validity of your manipulations in your main study as well.

7. For your collected sample, you mention fully completed and partially completed responses. I had to pull out a calculator to figure out how you got 577 participants, because you don't explain whether you kept the partially completed responses or deleted them. You need to clearly explain what respondents you used for your data analysis.

8. Your instrument section should be at the beginning of the method section, not the end.

9. Your instrument section just states “finally, they answered a series of attitudinal and learning-related questions to measure their level of understanding of key content covered in the film” – you need to state what each of these measures are and a citation for each one on where you got the measure from.

10. You need a section in your methods section that explain every outcome variable, control variable, and attention checking question that you measured in your survey. Each measure listed in that section should have a definition of the measure, an in-text citation for the source got the measure from, the items in each measure, whether the measure was a Likert or semantic differential scale, the number of items in the measure, and the Cronbach's alpha level. Each Cronbach's alpha should be above .7 and have more than three items to be considered a reliable measure. If it's below, provide rationalization for why you are still using the measure.

11. There is no power analysis or reasoning shown to establish why you chose to collect a sample size of n=577 participants. Without sample size reasoning, you have no way of knowing if your results are due to chance or are genuine.

A power analysis establishes an acceptable probability of making a type 1 or type 2 error for rejecting or failing to reject your null hypothesis for your experiment’s given circumstances.

Minor issues

1. There are statements in this paper that need in-text citations for support. For example, on pg. 3, the statement “One promising science communication approach that has gained increasing attention in recent years is cinematic-style 3D data visualizations” needs citations. There are parts throughout the lit review that need evidence either through a citation or your own personal reasoning.

2. Add in a model with all of your study's variables, and add in more tables for your data results (like your participant demographics). Then, just have the reader refer to them.

3. For your methods, you really only need the following sections: an intro after the Methods section title stating something like: "Study one was conducted using a single (with two levels: label and no label), between-subjects, factorial design via an online experiment." Then a Participants (talk about how you attained your sample, sampling method, sample size, power analysis, and reasoning as to way your sample and sampling method are appropriate for your study based on your variables used), Procedure and Stimulus, Measures, and Analytical Approach section.

--

Again, this is paper has a great idea in terms of extending knowledge in the field of science communication through data visualization. While this is not an exhaustive list, I tried to address the items that needed the most attention in this paper.

6. PLOS authors have the option to publish the peer review history of their article (what does this mean?). If published, this will include your full peer review and any attached files.

Reviewer #1: No

---

## [Author Response · Author response to Decision Letter 0]

2 Nov 2023

I sent an email to confirm that PLOSONE was not requiring a psychometric approach to quantitative social science, as the reviewer had indicated was mandatory. On this basis, I have adjusted the manuscript based on the feedback that is not specific to the psychometric tradition.

A detailed response to the reviewer comments has been provided in the 'response to reviewers' attachment, so I am not clear on what the point of this text box is. I have deleted Figure 1 to reduce complexity given PLOS ONE's administrative requirements.

---

## [Decision Letter · Decision Letter 1]

25 Dec 2023

PONE-D-23-23886R1Picture Perfect Science Communication: How Public Audiences Respond to Informational Labels in Cinematic-Style 3D Data VisualizationPLOS ONE

Dear Dr. Jensen,

Thank you for submitting your manuscript to PLOS ONE. After careful consideration, we feel that it has merit but does not fully meet PLOS ONE’s publication criteria as it currently stands. Therefore, we invite you to submit a revised version of the manuscript that addresses the points raised during the review process.

The paper has been re-examined by the previous reviewer. No substantial modifications are required, but just a number of editorial modifications.==============================

We look forward to receiving your revised manuscript.

Kind regards,

Maurizio Naldi

Academic Editor

PLOS ONE

Journal Requirements:

Reviewers' comments:

Reviewer's Responses to Questions

**Comments to the Author**

1. If the authors have adequately addressed your comments raised in a previous round of review and you feel that this manuscript is now acceptable for publication, you may indicate that here to bypass the “Comments to the Author” section, enter your conflict of interest statement in the “Confidential to Editor” section, and submit your "Accept" recommendation.

Reviewer #1: (No Response)

2. Is the manuscript technically sound, and do the data support the conclusions?

Reviewer #1: Yes

3. Has the statistical analysis been performed appropriately and rigorously? 

Reviewer #1: Yes

4. Have the authors made all data underlying the findings in their manuscript fully available?

Reviewer #1: Yes

5. Is the manuscript presented in an intelligible fashion and written in standard English?

Reviewer #1: Yes

6. Review Comments to the Author

Reviewer #1: I thank the authors for their tedious and thoughtful feedback on my comments. Each of my comments has been either addressed adequately, or the authors have provided ample reasoning for why items were kept as is in the manuscript.

I have one larger edit and a few small, minor edits for the authors to make, and then I am comfortable with accepting this manuscript for publication.

Major Edit:

1. You provide some great information and reasoning in your discussion section; however, your discussion section should dive into the “why” for your results a little more deeply. While many of your results were not significant (and you are of course not supposed to make inferences on non-significant results), you can still provide thoughtful reasoning as to why you think a relationship may not have occurred. For example, on page 22, you state:

“The three items used to evaluate the immersive nature of the videos – ease of mind, attention focus, and feeling as if in space – showed no statistically significant differences between the labeled and unlabeled videos.”

You could provide a sentence or two for each one (ease of mind, attention focus, and feeling as if in space) explaining why you think there was not a relationship between the variables that were tested. You can talk about the implications of these non-significant findings for this article’s area of research more deeply as well.

Many readers go straight to the discussion section of a research paper when looking for research to cite so this can help increase the number of times your article gets cited in future research.

Minor Edits:

1. There is an instance on page 18 in the second paragraph of that page where an in-text citation reads “Error! Reference source not found.” Just make sure instances like these are fixed in the manuscript or provide reasoning for why these are written this way.

2. On page 22, in the first paragraph of the Discussion section, add a “)” after 574 in this sentence: “A mean score of 80.74 out of 100 represented a generally positive sentiment about video quality (n=574.” Again, while this manuscript is well-written, just make sure any little grammatical instances like this one are corrected throughout the manuscript.

3. On page 23, there is a link to a blog post embedded in this sentence: “To do this, we believe in taking an evidence-based communication approach, with audience research in a project to systematically test different design options.” Make sure to remove instances like this throughout the manuscript.

4. Some of the tables go off the page. This may be fixed when the manuscript is formatted after being accepted for publication; however, you could adjust the dimensions of these to fit the page.

--

Lastly, I know the R&R process for a journal article publication can be rigorous and stressful, but (as you likely know) it is all to create a higher-quality article for the journal and the authors. I am excited to see this article get published.

7. PLOS authors have the option to publish the peer review history of their article (what does this mean?). If published, this will include your full peer review and any attached files.

Reviewer #1: No

---

## [Editor Report · Decision Letter 2]

17 Jan 2024

PONE-D-23-23886R2Picture Perfect Science Communication: How Public Audiences Respond to Informational Labels in Cinematic-Style 3D Data VisualizationPLOS ONE

Dear Dr. Jensen,

Thank you for submitting your manuscript to PLOS ONE. After careful consideration, we feel that it has merit but does not fully meet PLOS ONE’s publication criteria as it currently stands. Therefore, we invite you to submit a revised version of the manuscript that addresses the points raised during the review process.

Please comply with all the revision requests. They do not imply further research work, but may require extensive editorial work and argumentation.==============================

We look forward to receiving your revised manuscript.

Kind regards,

Maurizio Naldi

Academic Editor

PLOS ONE
---

## [Editor Report · Decision Letter 3]

11 Jul 2024

Picture Perfect Science Communication: How Public Audiences Respond to Informational Labels in Cinematic-Style 3D Data Visualization

PONE-D-23-23886R3

Dear Dr. Jensen,

We’re pleased to inform you that your manuscript has been judged scientifically suitable for publication and will be formally accepted for publication once it meets all outstanding technical requirements.

Kind regards,

Maurizio Naldi

Academic Editor

PLOS ONE
---

## [Editor Report · Acceptance letter]

7 Oct 2024

PONE-D-23-23886R3 

PLOS ONE

Dear Dr. Jensen, 

I'm pleased to inform you that your manuscript has been deemed suitable for publication in PLOS ONE. Congratulations! Your manuscript is now being handed over to our production team.

Kind regards, 

on behalf of

Professor Maurizio Naldi 

Academic Editor

PLOS ONE